# SAM-Geo3D: A Geometrical Method to Extend SAM to 3D

**Jikai Zhang**[*][1]                                          JIKAI.ZHANG@DUKE.EDU

[1] *Department of Electrical and Computer Engineering, Duke University, Durham, NC, 27708, USA*

**Zafer Yildiz**[*][2]                                          ZAFER.YILDIZ@DUKE.EDU

[2] *Department of Radiology, Duke University, Durham, NC, 27708, USA*

**Hanxue Gu**[1]                                                HANXUE.GU@DUKE.EDU

**Haoyu Dong**[1]                                               HAOYU.DONG@DUKE.EDU

**Maciej A. Mazurowski**[1,2,3,4]                               MACIEJ.MAZUROWSKI@DUKE.EDU

[3] *Department of Computer Science, Duke University, Durham, NC, 27708, USA*

[4] *Department of Biostatistics & Bioinformatics, Duke University, Durham, NC, 27708, USA*

## Abstract

Segment Anything Model (SAM) offers a promising approach for image segmentation tasks. However, SAM works in 2D, making it less useful when segmenting cross-sectional images, such as MRIs. To address this, we proposed SAM-Geo3D, a geometrical method that extends SAM into the 3D. Provided a few prompt points on a target object, SAM-Geo3D segments the object through all slices in 3D without onerous deep-learning-based training. We validated SAM-Geo3D on five knee MRI volumes. Results showed that SAM-Geo3D outperforms SAM when using the same, limited number of input prompt points.

**Keywords:** Image Segmentation, Segment Anything Model, 3D SAM

## 1. Introduction

Segment Anything Model (SAM) is a seminal foundation model (Kirillov et al., 2023) designed to segment any object of interest in 2D images based on given prompts. However, it cannot produce as promising segmentation results as in the natural image domain when applied to the medical image domain (Mazurowski et al., 2023). One potential reason for this weakness is that SAM is primarily designed for 2D inputs and thus cannot utilize the context relationship between consecutive slices in 3D medical images, which can be useful to track continuous objects through different slices. To eliminate this deficiency, we aim to extend SAM to 3D inputs. Even though there are some existing works (Wang et al., 2023; Li et al., 2023; Chen et al., 2023), they require additional **training** of SAM on 3D datasets, which would require additional resources and potentially lead to overfitting.

In this work, we proposed SAM-Geo3D, a geometrical method that extends SAM into 3D. SAM-Geo3D only requires a few prompt points and can segment the entire object, including parts within slices that do not have prompts provided. Our method may alleviate human labor and does not require additional data and training, presenting a promising avenue for further research into SAM's segmentation capabilities.

---

[*] Contributed equally

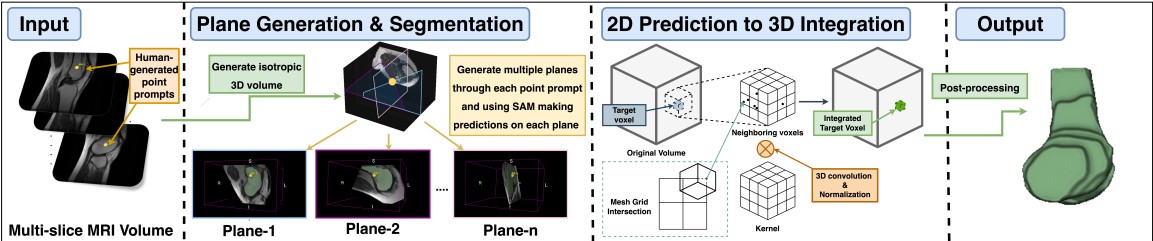

Figure 1: Pipeline of SAM-Geo3D. We randomly sampled planes and obtained plane intersections with the 3D volume. Each plane intersection is modified as a 2D regular grid. Grid points are filled with corresponding voxel values in 3D. We applied SAM to generate segmentation on plane images and assigned mask values on grid points. Integration algorithm summarizes a target voxel using the information in neighboring voxels.

## 2. Method

SAM-Geo3D takes user-input prompt points on the object of interest from the input volume as the input. The output of SAM-Geo3D is the full 3D segmentation mask of all objects in the input volume. SAM-Geo3D consists of two main modules: *random plane generation* and *integration*, shown in Fig. 1.

**Prompt Generation**   In this work, we validated SAM-Geo3D with a specific task in 3D bone segmentation on knee MRIs. We utilized a connected component algorithm on the annotated bones to represent 3D bone objects. Then, we randomly sampled five prompt points per 3D component as the simulated user-input prompts.

**Plane Generation**   By leveraging the planes that contain the user-input prompt point and intersect with the 3D volume, we aim to expand SAM into the 3D manner. First, we applied nearest-neighbor interpolation on original volumes to acquire isotropic unit voxels. Starting at each prompt point input, we randomly initialized 100 normal vectors to generate 100 planes. To extract the plane intersection, for each plane $\mathbf{P}$, we created a 2D coordinate system with $\mathbf{x}$ and $\mathbf{y}$ axes both orthogonal to each other and the normal vector of $\mathbf{P}$. The origin of the new coordinate system is the prompt point. To ensure we included all intersection points between $\mathbf{P}$ and volume, we started from the origin and moved in 4 directions in 3D space $(-\mathbf{x}, +\mathbf{x}, -\mathbf{y}, +\mathbf{y})$ until we reached the boundary of the 3D volume. As we moved in 4 directions, we created a 2D regular grid with unit grid length and formed a mapping between the 2D coordinates on the grid $(p_x, p_y)$ and the corresponding 3D coordinates $(v_x, v_y, v_z)$ in the volume. Once we have completed the mapping, we fetched the voxel value for each pixel on the plane $(Value_{(p_x, p_y)} = Value_{(Floor(v_x), Floor(v_y), Floor(v_z))})$. Pixels outside the volume were zero-padded. Finally, SAM was applied to each 2D plane image to generate random plane segmentations based on initial prompt point inputs.

**Integration algorithm**   For each 3D voxel, we collected random plane segmentations. If a voxel location has multiple planes intersected, we recorded all the results. If there is no plane intersected, we recorded NULL. A fixed $3 \times 3 \times 3$ Gaussian kernel was then applied at each voxel location to summarize the non-null segmentation results.

**Post-processing** First, we applied a thresholding strategy to binarize the integration output. We tuned the thresholds of $0.5, 0.6, 0.7, 0.8$ on the development volume. Next, we applied a connected component algorithm to identify the largest piece of the component in 3D. Finally, we compressed the results from interpolated space to the original space by averaging and binarizing outputs on identical, interpolated slices.

## 3. Experiments and Discussion

**Dataset** Our dataset consists of six knee MRI volumes collected internally. Full-volume annotations of tibia, fibula, patella, and femur were collected. Despite no training is required for SAM-Geo3D, we randomly selected one volume for the algorithm development and hyperparameter selection. We evaluated SAM-Geo3D on the remaining five volumes.

**Evaluation** For evaluation, we gathered results from all components, applied boolean additions to obtain one final binarized mask per volume, and reported the following metrics. $GlobalMean$ takes the average of dice coefficients on every slice. $LocalMean$ takes the average of dice coefficients on slices only with initial prompt points. We provided results for SAM with (1) the same number of prompt point inputs as SAM-Geo3D and (2) extra prompt points so that every object on every slice has at least one prompt point.

**Results** Table 1 compares global/local performance between SAM and SAM-Geo3D. On average, 55 prompt points were generated to cover all objects on every slice for SAM. When using the same 19 prompt point inputs, SAM-Geo3D outperforms SAM on Dice-G ($+0.378$, $P < 0.01$) and Dice-L ($+0.252$, $P < 0.01$). When using almost three times more prompt points, SAM performs comparably to SAM-Geo3D (Dice-G: $-0.138$, $P = 0.05$; Dice-L: $-0.022$, $P = 0.3$). Figure 2 shows a set of SAM-Geo3D results on a selected slice compared to SAM. SAM-Geo3D provides masks on objects without prompt points.

**Conclusion and Future Work** SAM-Geo3D provides a geometrical way to use SAM without onerous fine-tuning tasks in deep learning. With a limited number of prompt points, SAM-Geo3D outperforms SAM in providing reliable segmentation masks on objects without prompt point inputs. For future work, we will increase the scale of our dataset, experiment with multiple numbers of prompt points and planes, improve the quality of segmentation outputs by smoothing/filling the mask, and improve the algorithm efficiency.

| Model | # PP | Dice-G | Dice-L |
|---|---|---|---|
| SAM | 55 | 0.790 | 0.818 |
| SAM | 19 | 0.274 | 0.544 |
| 3DSAM | 19 | 0.652 | 0.796 |

Table 1: The performance of SAM and SAM-Geo3D. # PP refers to the number of prompt points. Dice-G refers to averaging dice core according to GlobalMean, and -L refers to LocalMean.

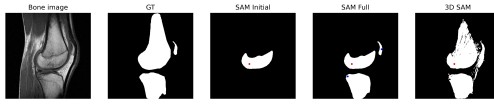

Figure 2: Visualization of SAM and SAM-Geo3D on one example slice. GT is the ground truth annotation. Initial prompts are colored in red. Extra prompt points are colored in blue.

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
