# OpenReview forum: "SAM-Geo3D: A Geometrical Method to Extend SAM to 3D"
_MIDL.io/2024/Short_Papers — MIDL 2024 Short Papers_

### Official Review · Reviewer_xCAQ · 2024-04-22

**Confidence:** 5
**Final Rating:** 3.5

**Review:**

Summary:

Pros:

1.	The paper proposed SAM-Geo3D, a novel geometrical method that extends SAM into the 3D manner, which eliminates the deficiency of SAM’s inability while being applied to 3D medical image segmentation and avoids onerous deep learning training;
2.	The performance improvement is very clear.

Cons:
1.	The quality and clarity of the paper is very poor. E.g., (a) what does “LocalMean” metric mean, what is the range of locality? (b) what does “2D mesh grid” mean, is it regular or irregular? (3) Is the 3x3x3 Gaussian kernel learnable with only one training sample?
2.	The paper lacks the conclusion section;
3.	The paper conducts experiment on the internal dataset with very limited samples, which cannot verify the robustness and generalizability;
4.	The application of multiple thresholds is inefficient during the post-processing step.
5.	The authors should articulate how to compress the results from interpolated space to the original space.

---

### Decision · Program_Chairs · 2024-04-26

Accept